# MAF1 is a predictive biomarker in HER2 positive breast cancer

**Stephanie Cabarcas-Petroski**[1]*, **Gabriella Olshefsky**[2], **Laura Schramm**[3]

**1** Department of Biology, Pennsylvania State University, Beaver Campus, Monaca, PA, United States of America, **2** Garden City High School, Garden City, NY, United States of America, **3** Department of Biology, St. John's University, Queens, NY, United States of America

* smc1088@psu.edu

**Data Availability Statement:** The data can be found in public respositories. The URLs can be found embedded in the paper with accession numbers/DOIs in Table 1. I have also included links below to where the data can be found: https://pdc.

## Abstract

RNA polymerase III transcription is pivotal in regulating cellular growth and frequently deregulated in various cancers. MAF1 negatively regulates RNA polymerase III transcription. Currently, it is unclear if MAF1 is universally deregulated in human cancers. Recently, MAF1 expression has been demonstrated to be altered in colorectal and liver carcinomas and Luminal B breast cancers. In this study, we analyzed clinical breast cancer datasets to determine if MAF1 alterations correlate with clinical outcomes in HER2-positive breast cancer. Using various bioinformatics tools, we screened breast cancer datasets for alterations in MAF1 expression. We report that MAF1 is amplified in 39% of all breast cancer subtypes, and the observed amplification co-occurs with MYC. MAF1 amplification correlated with increased methylation of the MAF1 promoter and MAF1 protein expression is significantly decreased in luminal, HER2-positive, and TNBC breast cancer subtypes. MAF1 protein expression is also significantly reduced in stage 2 and 3 breast cancer compared to normal and significantly decreased in all breast cancer patients, regardless of race and age. In SKBR3 and BT474 breast cancer cell lines treated with anti-HER2 therapies, MAF1 mRNA expression is significantly increased. In HER2-positive breast cancer patients, MAF1 expression significantly increases and correlates with five years of relapse-free survival in response to trastuzumab treatment, suggesting MAF1 is a predictive biomarker in breast cancer. These data suggest a role for MAF1 alterations in HER2-positive breast cancer. More extensive studies are warranted to determine if MAF1 serves as a predictive and prognostic biomarker in breast cancer.

## Introduction

Breast cancer in the United States (U.S.) accounts for 31% of new cancer diagnoses in women [1]. The most commonly diagnosed breast cancer is invasive breast cancer (IBC), with approximately 1 in 8 women (13%) diagnosed with IBC and 1 in 39 (3%) deaths from IBC [2]. Breast cancers are classified by site and whether the breast cancer is invasive or non-invasive [3]. In 2023, 297,790 new cases of IBC and 51,400 cases of non-IBC diagnoses are anticipated [1, 2]. The most commonly diagnosed breast cancers are invasive ductal carcinoma (IDC), ductal

cancer.gov/pdc/ http://tcga-data.nci.nih.gov/tcga/ https://www.ncbi.nlm.nih.gov/geo/query/acc.cgi?acc=GSE129254 https://www.ncbi.nlm.nih.gov/geo/query/acc.cgi?acc=GSE158969 https://www.ncbi.nlm.nih.gov/geo/query/acc.cgi?acc=GSE157383 https://www.ncbi.nlm.nih.gov/geo/query/acc.cgi?acc=GSE99060 https://www.rocplot.org/.

**Funding:** The author(s) received no specific funding for this work.

**Competing interests:** The authors have declared that no competing interests exist.

carcinoma in situ (DCIS), and invasive lobular carcinoma (ILC) [3]. IDC and ILC account for 90% of IBC, whereas DCIS is the most common non-invasive breast cancer diagnosis [4].

Genetic and epigenetic alterations often lead to deregulated cell proliferation in human cancers. Cell proliferation is partially regulated by three eukaryotic RNA polymerases (pol) [5]. RNA pol I transcribes ribosomal RNA; RNA pol II transcribes mRNA-encoding proteins and select untranslated RNAs involved in RNA metabolism, whereas RNA pol III transcribes untranslated RNA molecules [5]. The role of these RNA pol's, specifically RNA pol I and III, are implicated as critical regulators of the biosynthetic capacity of a cell.

TFIIIB, established as deregulated in human cancers, is required to initiate RNA pol III transcription accurately [5–15]. Two forms of TFIIIB are defined in higher eukaryotes [11, 16]. Accurate RNA pol III transcription from gene-external promoters requires a TFIIIB complex containing BDP1, TBP, and BRF2 [11, 16]. A related TFIIIB complex is necessary for gene-internal RNA pol III promoters containing BDP1, TBP, and BRF1 [11, 16]. TFIIIB and RNA pol III transcription activities are inhibited by tumor suppressors, including p53 [17, 18], PTEN [19–21], BRCA1 [22], the retinoblastoma protein (Rb) [18], and the Rb family members p130 and p107 [23]. The oncogenes MAP kinase ERK and MYC [12, 18] stimulate TFIIIB activity *in vitro*. Specifically, the TFIIIB subunits TBP [24–26], BRF1 [6, 27–29], BRF2 [6, 7, 30–37], and BDP1 [8, 9, 12, 21, 38, 39] are altered in a variety of human cancers, including breast, blood, colorectal, cervical, esophageal, liver, lung, prostate, and skin cancers.

MAF1, initially identified in yeast in response to cellular stress and nutritional deprivation, is a general negative regulator of RNA pol III transcription [40, 41]. MAF1 is conserved in eukaryotes, including humans [42, 43]. During cellular stress or nutrient starvation, coinciding with the inactivation of mTORC1 kinase, human MAF1 is dephosphorylated, impairing the recruitment of RNA pol III machinery [44].

PI3K-AKT-mTOR activation is frequently observed in human cancers [45], and is a target for therapeutic design [46]. Specifically, in hepatocellular carcinoma (HCC) patients, decreased MAF1 expression correlated with poor prognosis [47]. In colorectal cancer, increased MAF1 expression is associated with metastasis and poor prognosis [48]. In a sample size of 192 luminal B breast cancer patients, MAF1 copy number is amplified [49]. To date, the analysis of MAF1 expression in human cancers has been limited and has not been extensively studied. MYC, located on chromosome 8q24.21, has been identified as a frequent amplifier in HER2+ breast cancer. MAF1, as stated, is a negative regulator of RNA pol III transcription, and is located on chromosome 8q24.3, in close proximity to MYC.

Taken together, we sought to determine if MAF1 could also be implicated in HER2+ breast cancer [50]. This study aims to determine if MAF1 expression correlates with clinical outcomes in HER2-positive breast cancer. Herein, we report that MAF1 is amplified in 39% of all breast cancer sub-types, and the observed amplification co-occurs with MYC. MAF1 amplification correlated with increased methylation of the MAF1 promoter. MAF1 protein expression is significantly decreased in luminal, HER2-positive, and TNBC breast cancer subtypes. In stage 2 and 3 breast cancer, MAF1 protein expression is significantly reduced compared to normal and protein expression is significantly decreased in all breast cancer patients, independent of race and age. Recently, it was demonstrated that the amplification of HER2 and deregulation of MYC accelerated tumorigenesis, metastasis, and lethality in breast cancer [50]. Hence, we analyzed MAF1 expression in response to anti-HER2 therapies in breast cancer cell lines and patients. MAF1 mRNA expression significantly increases in SKBR3 and BT474 breast cancer cell lines treated with anti-HER2 therapies. Lastly, we also report increased MAF1 expression correlating with an increased five year relapse-free survival response to trastuzumab treatment, suggesting MAF1 is a predictive biomarker in breast cancer.

**Table 1. Datasets analyzed.**

| Dataset | Study Description and Link to Dataset | Ref |
|---------|---------------------------------------|-----|
| CPTAC | The proteogenomic landscape of breast cancer dataset generated by the Clinical Proteomic Tumor Analysis Consortium (CPTAC) includes 122 samples publicly available through the CPTAC data portal https://cptac-data-portal and at the Proteomic Data Commons (https://pdc.cancer.gov/pdc/). The proteogenomic landscape of breast cancer dataset was accessed and analyzed via the cBioPortal, January 2021 –March 2023. | [51] |
| TCGA | The Cancer Genome Atlas Program (TCGA)(http://tcga-data.nci.nih.gov/tcga/) contains data for over 20,000 primary cancer and normal samples spanning 33 cancer types. In the current study, we utilized the invasive ductal carcinoma datasets ($n = 1602$), accessed January 2021 –March 2023. | [52–54] |
| GSE129254 | Gene expression profiling was examined by Human HT-12 v4.0 Expression BeadChip arrays in SKBR3 and BT474 cells treated with HER2 inhibitor lapatinib ($n = 18$), accessed January 2021 –March 2023. | [55] |
| GSE158969 | Illumina HiSeq paired-end RNA sequencing of BT474 cells treated with trastuzumab ($n = 30$), accessed January 2021 –March 2023. | [56] |
| GSE157383 | Illumina HiSeq single-end RNA sequencing of human breast cancer cell lines treated with abemaciclib ($n = 22$), accessed January 2021 –March 2023. | [57] |
| GSE99060 | Torrent Suite (Thermo Fisher) single-end RNA sequencing of human breast cancer cell lines treated with abemaciclib ($n = 18$), accessed January 2021 –March 2023. | [58] |

## Materials and methods

### Datasets analyzed

The accession numbers for all datasets analyzed in this study can be found in Table 1. The current study did not require IRB or ethics committee approval as we performed *in silico* analysis for publicly available datasets. We did not collect human samples (tissues, cells, fluids) or communicate with human participants. The data used for analysis were retrieved from publicly accessible datasets. We refer readers to prior publications detailing methods [7–9, 38].

### cBioPortal analysis of MAF1

We queried for MAF1 alteration in breast cancer using the proteogenomic landscape of breast cancer dataset ($n = 122$) generated by the Clinical Proteomic Tumor Analysis Consortium (CPTAC) [51] housed in the cBioPortal for Cancer Genomics, an open-source multi-cancer genomics and clinical dataset analysis platform [59] (Table 1). *P*-values are derived from the Log Rank test, and the *q*-values are derived from the Benjamini-Hochberg False Discovery Rate (FDR) correction procedure.

### University of Alabama at Birmingham Cancer (UALCAN) data analyses

The University of Alabama at Birmingham Cancer (UALCAN) data analysis portal [60] was accessed from June 2022 through March 2023 to analyze the TCGA breast cancer dataset [52–54] for MAF1 mRNA expression and MAF1 promoter methylation. The portal contains protein expression datasets [61] to determine MAF1 protein expression in HER2-positive breast cancer.

### GEO2R platform microarray analysis

Within the GEO2R platform [62], the limma (Linear Models for Microarray Analysis) R package was used to analyze the publicly available expression array GSE129254 dataset [55] for differentially expressed genes (DEGs) in response to anti-HER2 therapies. A log transformation was performed on the data. The Benjamini & Hochberg false discovery rate (FDR) method

**Table 2. MAF1 and TFIIIB are differentially expressed in response to anti-HER2 therapies.**

| GSE129254 | | | | |
|---|---|---|---|---|
| **Cell line/drug** | **adj. *p*-value** | **_p_-value** | **logFC** | **gene symbol** |
| BT474/ lapatinib | 0.026 | 0.002 | 0.731 | MAF1 |
| SKBR3/ lapatinib | 0.086 | 0.002 | 0.701 | MAF1 |
| BT474/ lapatinib | 0.078 | 0.015 | -0.434 | BRF2 |
| SKBR3/lapatinib | 0.587 | 0.310 | 0.155 | BRF2 |
| BT474/ lapatinib | 0.235 | 0.085 | -0.294 | BRF1 |
| SKBR3/lapatinib | 0.613 | 0.341 | 0.396 | BRF1 |
| BT474/ lapatinib | 0.434 | 0.234 | 0.545 | BDP1 |
| GSE158969 | | | | |
| **Cell line/drug** | **adj. *p*-value** | **_p_-value** | **logFC** | **gene symbol** |
| BT474/trastuzumab | $4.51 \times 10^{-5}$ | $8.00 \times 10^{-6}$ | 0.231 | MAF1 |
| BT474/trastuzumab | $1.28 \times 10^{-6}$ | $1.60 \times 10^{-7}$ | 0.372 | BRF2 |
| BT474/trastuzumab | 0.352 | 0.275 | -0.034 | BRF1 |
| BT474/trastuzumab | 0.057 | 0.035 | 0.421 | BDP1 |

(adj. *p*-value) was employed to correct for false positive results (*p*-values), and the results of our genes of interest are presented in Table 2. GEO2R was accessed from June 2022 –March 2023.

**Galaxy analyses.** The Galaxy platform [63] was utilized to identify differentially expressed genes in the GSE158969 dataset [56] using limma and edgeR DEGs tools. Volcano plots were generated in Galaxy using ggplot2 to visualize DEGs within the GSE129254 [55] and GSE158969 datasets [56].

## ROC plotter analyses

The ROC Plotter portal [64] is a receiver operating characteristic (ROC) tool for meta-analysis-based discovery and validation of survival biomarkers [64]. Based on their clinical characteristics, breast cancer dataset samples are divided into responder and nonresponder groups. The groups were further analyzed using the Mann-Whitney and ROC tests in the R statistical environment using Bioconductor libraries [64]. The cutoff for *p* values is $p < 0.05$, and only results with a 5% false discovery rate (FDR) were considered significant [64]. We queried the ROC plotter platform to analyze MAF1 (probe 222998_at) expression in response to anti-HER2 therapies and clinical outcomes, specifically relapse-free survival status for five years, accessed January 2022 –March 2023.

## Results

### MAF1 and MYC amplifications co-occur in breast cancer

This study aims to determine if MAF1 expression is altered in breast cancer subclasses and if MAF1 alterations correlate with clinical outcomes in breast cancer sub-types. Using the cBio-Portal [59, 65], we queried the proteogenomic landscape of breast cancer dataset [51] to determine if MAF1 expression was altered in breast cancer. Fig 1A demonstrates MAF1 amplification in 39% of invasive breast cancers (n = 122). We further analyzed the dataset for known biomarkers in breast cancer, including estrogen receptor alpha (ESR1), the progesterone receptor (PGR) and HER2 (ERBB2) [66]. In Fig 1A, we demonstrate that these common breast cancer biomarkers used in the clinic are amplified, including ESR1(5%), PGR(6%), and HER2 (ERBB2)(15%). In breast cancer, it has been demonstrated that chromosome 8q24 is frequently amplified [67]. MYC is located on chromosome 8q24.21, MAF1 is on 8q24.3. We

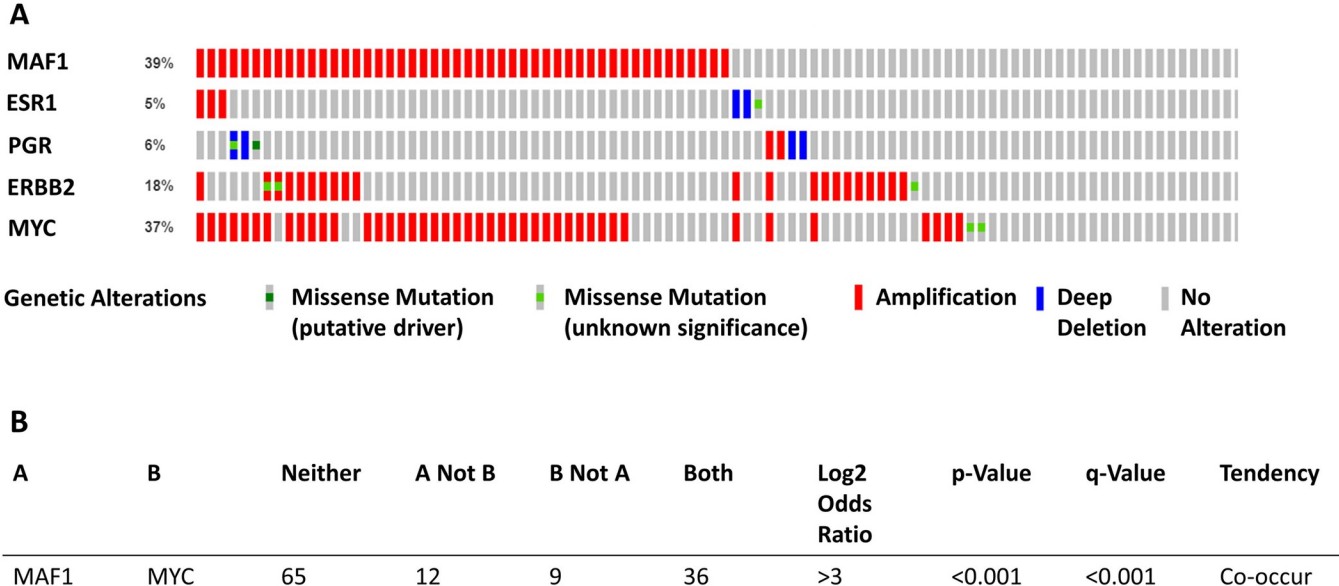

**Fig 1. MAF1 amplification frequently occurs in breast cancer patients.** The proteogenomic landscape of breast cancer dataset generated by the Clinical Proteomic Tumor Analysis Consortium (CPTAC) [51] was analyzed using cBioPortal [59] We generated an OncoPrint (**A**) of alterations of MAF1, MYC, PGR, and ERBB2. MAF and MYC amplifications significantly co-occur (p <0.001) (**B**) in the CPTAC dataset. Genetic alteration subtypes are noted in (**A**). P-values (one-sided Fisher Exact Test) and q-values (Benjamini-Hochberg FDR correction procedure) are presented (**B**).

analyzed MAF1 and MYC for frequency of co-occurrence and mutual exclusivity. Our analysis shows that MYC and MAF1 alterations significantly co-occur in breast cancer ($q$ = 0.001), Fig 1B. Our results are in agreement with a transcriptome analysis of HER2-positive breast cancer tumors (n = 99) that identifies MYC (19%) and MAF1 (6%) as frequently amplified in HER2--positive cancer [68].

## MAF1 protein expression is significantly decreased in luminal, HER2-positive, and TNBC breast cancer subtypes

Fig 1 demonstrates the amplification of MAF1 in breast cancer. However, it is established that not all genetic amplifications lead to functional changes. We wanted to determine if the observed MAF1 amplifications correlated with mRNA and protein expression changes in breast cancer datasets. We utilized the University of Alabama at Birmingham Cancer data analysis portal (UALCAN) [60] and queried the TCGA breast cancer dataset [52–54] to correlate MAF1 amplification with changes to MAF1 mRNA expression. MAF1 mRNA expression is significantly altered in luminal ($p$ = 1.68 x $10^{-5}$) and triple-negative breast cancer (TNBC) ($p$ = 2.01 x $10^{-7}$) but not in HER2-positive breast cancer, Fig 2A. The MAF1 promoter is significantly methylated in luminal ($p$ < 1 x$10^{-12}$), HER2-positive ($p$ = 2.26 x $10^{-3}$), and TNBC ($p$ = 5.26 x $10^{-12}$) subclasses of breast cancers, Fig 2B. Within UALCAN, we queried the Clinical Proteomic Tumor Analysis Consortium (CPTAC) and the International Cancer Proteogenome Consortium (ICPC) datasets [51] to analyze MAF1 expression. MAF1 protein expression is significantly decreased in luminal ($p$ = 2.40 x $10^{-8}$), HER2-positive ($p$ = 4.66 x $10^{-5}$), and TNBC ($p$ = 2.47 x $10^{-4}$), Fig 2C. Furthermore, an analysis of MYC demonstrated that decreased MYC protein expression occurs for both luminal ($p$ = 9.14 x $10^{-3}$) and HER2--positive ($p$ = 9.03 x $10^{-2}$) breast cancers, Fig 2D. As a control, we examined HER2 protein expression in the major subclasses of breast cancer, demonstrating no significant change in HER2 protein expression in HER2-positive breast cancer compared to normal, Fig 2E.

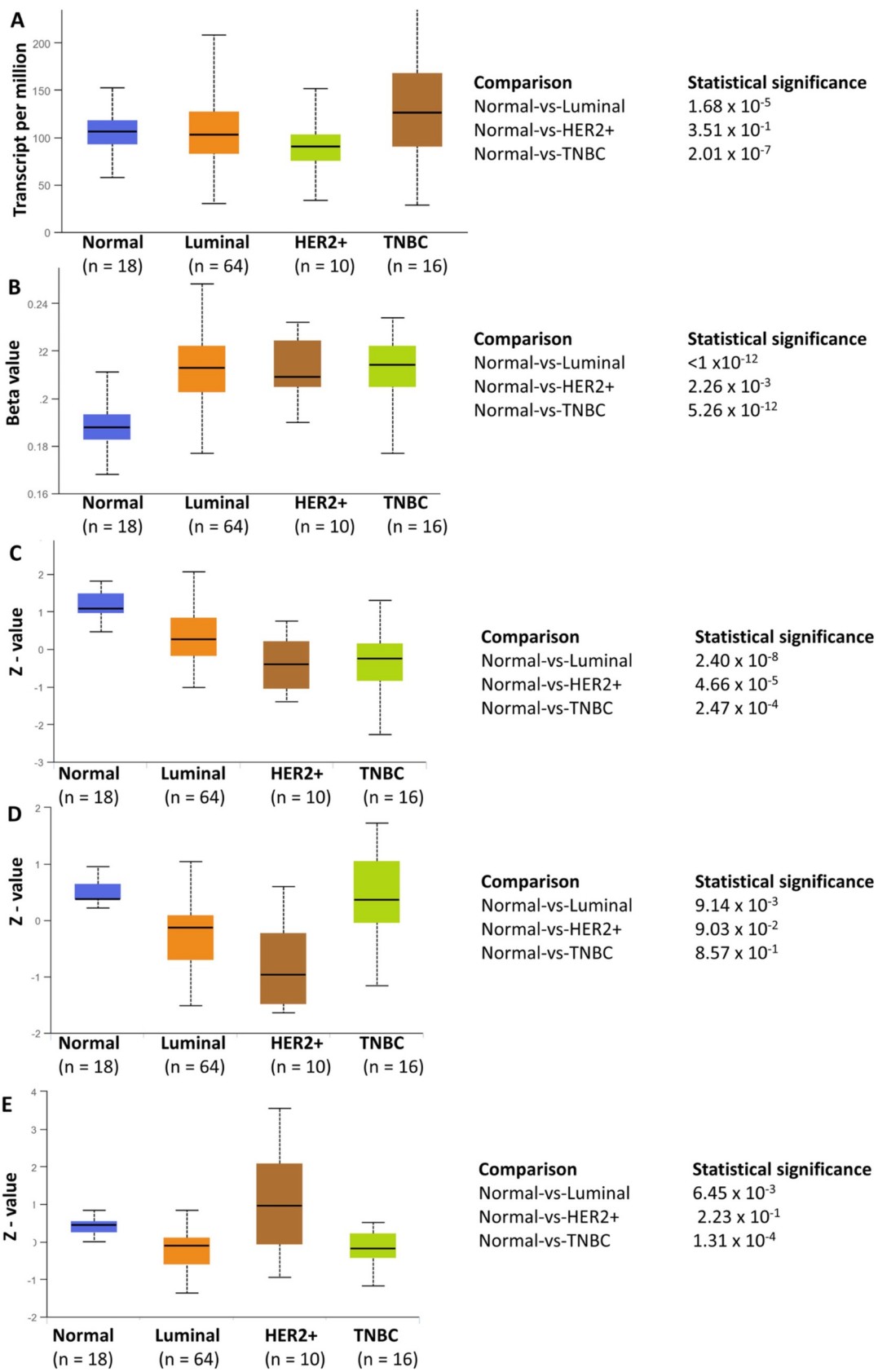

**Fig 2. MAF1 protein expression is significantly decreased in luminal, HER2-positive, and TNBC breast cancer subtypes.**
Using the University of Alabama at Birmingham Cancer data analysis portal (UALCAN) [60], we queried the TCGA breast cancer dataset [52–54] to determine MAF1 mRNA expression (**A**) and MAF1 promoter methylation in subclasses of breast cancer (**B**). UALCAN [60] provides a protein expression analysis option using data from Clinical Proteomic Tumor Analysis Consortium (CPTAC) and the International Cancer Proteogenome Consortium (ICPC) datasets [51]. MAF1 protein expression across breast cancer subtypes is depicted in (**C**), as compared to protein expression of MYC (**D**) and HER2 (**E**). The z-values presented are standard deviations of the median across samples, and log2 values were first normalized within each sample profile, then normalized across samples. Sample numbers and *p*-values are indicated.

## MAF1 protein expression is significantly decreased in stage 2 and 3 breast cancer and common histological types of breast cancer

In Fig 1, we demonstrate significant amplification and co-occurrence of MAF1 and MYC in breast cancer. Fig 2 demonstrates that MAF1 protein expression is substantially decreased in luminal, TNBC, and HER2-positive breast cancer. We then queried the UALCAN [60] proteo-genomic breast cancer patients' samples from the CPTAC and the International Cancer Proteogenome Consortium (ICPC) datasets [51] to analyze MAF1 protein expression by stage and histology sub-types in breast cancer. This analysis demonstrated that MAF1 protein expression is significantly decreased in stage 2 ($p = 7.35 \times 10^{-12}$) and stage 3 ($p = 1.25 \times 10^{-6}$) of breast cancer, compared to normal, Fig 3A. MAF1 protein expression is most significantly decreased in stage 2 breast cancer, a localized stage with the most favorable survival outcomes [1, 2]. In stage 3, a regional stage with lower survival rates, MAF1 protein expression is also significantly decreased [1, 2]. The MAF1 protein expression profile for stage 2 ($p = 7.35 \times 10^{-12}$). Next, we sought to determine if MAF1 protein expression correlates with breast cancer histology, Fig 3B. MAF1 protein expression is significantly decreased in IDC ($p = 2.07 \times 10^{-11}$) and ILC ($p = 3.59 \times 10^{-3}$), the first and second most common form of breast cancer, respectively [1, 2].

## MAF1 protein expression by patient race and age

We reviewed the available metadata for the samples (n = 99) utilized in the transcriptome analysis of HER2-positive breast cancer samples [68] and noted limited ethnic diversity. Of the 99 HER2-positive breast cancer patient samples sequenced, nine patients did not have ethnicity data, 89 identified as White/Caucasian, and one as Black/African [68]. Thus, using the CPTAC and ICPC datasets available in UALCAN, which contains larger sample sizes from more diverse populations, we sought to determine if MAF1 expression is altered in breast cancer patients by ethnicity and age. MAF1 protein expression decreased across the ethnicities tested, Fig 4A, with significant decreases in MAF1 protein in Caucasians ($p = 3.60 \times 10^{-10}$) (n = 80), Blacks/Africans ($p = 8.04 \times 10^{-6}$) (n = 18), and Asians ($p = 1.36 \times 10^{-6}$) (n = 20) compared to normal. Subsequently, we analyzed MAF1 protein expression and correlation with age in breast cancer, Fig 4B. Compared to normal, MAF1 protein expression decreased in breast cancer patients aged 21–80, Fig 4B, with the most significant decrease in patients aged 41–60 yrs ($p = 1.73 \times 10^{-10}$) and 61–80 yrs ($p = 6.34 \times 10^{-10}$). These data suggest that MAF1 protein expression in breast cancer is age-independent. Our results agree with previous studies indicating no significant association between patient age and HER2 or progesterone receptor protein expression in breast cancer [69].

## MAF1 mRNA expression is significantly increased in SKBR3 and BT474 breast cancer cell lines treated with anti-HER2 therapies

The decrease of MAF1 RNA and protein expression in breast cancer (Fig 2), including HER2-positive breast cancer (Fig 2C), prompted us to investigate whether anti-HER2 therapies

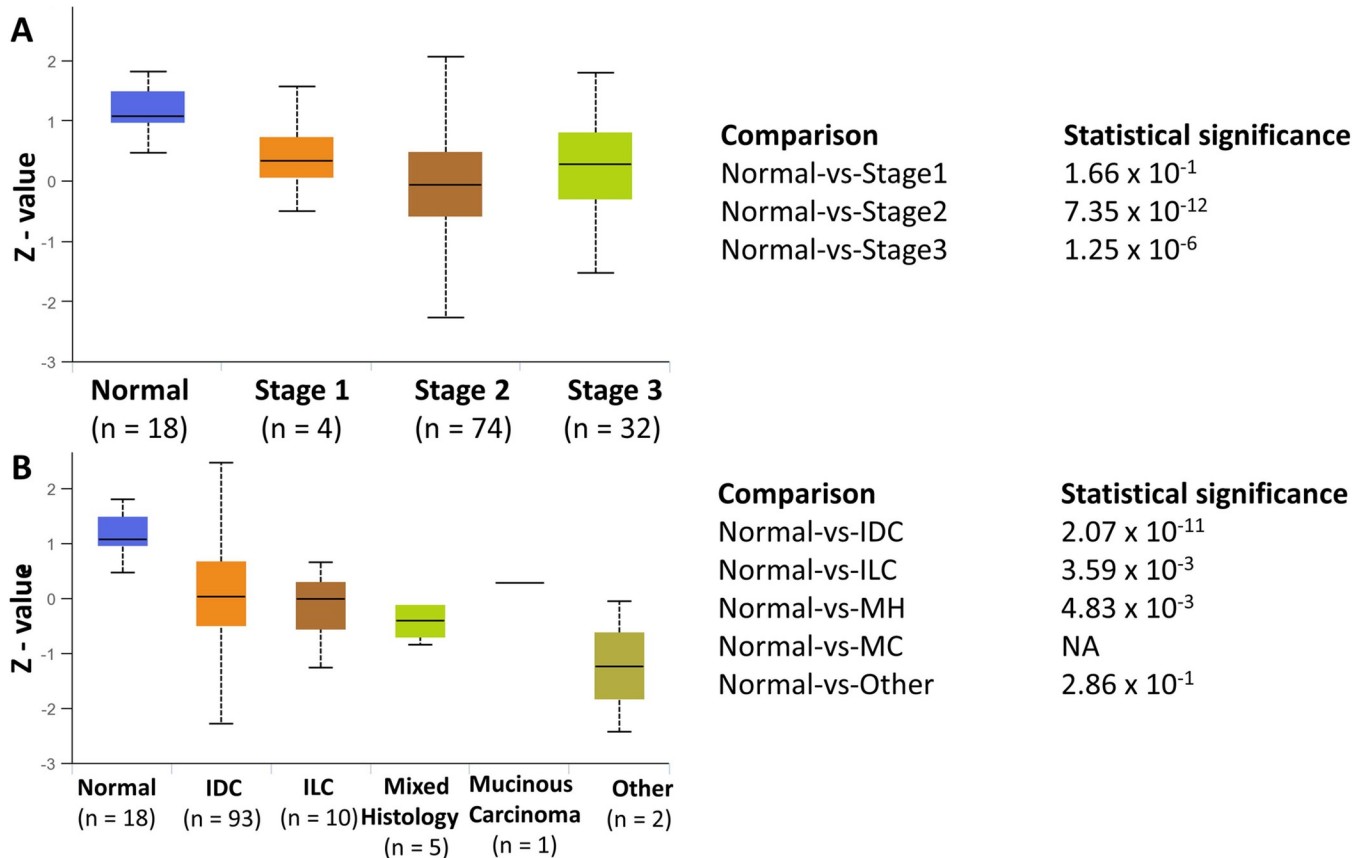

**Fig 3. MAF1 protein expression by stage and tumor histology.** We queried for MAF1 protein expression data by breast cancer stage (**A**) and tumor histology (**B**) using the Clinical Proteomic Tumor Analysis Consortium (CPTAC) and the International Cancer Proteogenome Consortium (ICPC) datasets [51], using the University of Alabama at Birmingham Cancer data analysis portal (UALCAN) [60]. The z-values presented are standard deviations of the median across samples, and log2 values were first normalized within each sample profile, then normalized across samples. Sample numbers are indicated; *p*-values are noted.

regulate MAF1 expression. Using the GEO2R platform [62] and the limma (Linear Models for Microarray Analysis) R package, we analyzed the expression array GSE129254 [55] dataset for differentially expressed genes (DEGs) comparing HER2-positive SKBR3 and BT474 breast cancer cell lines treated with DMSO or the selective ATP-competitive tyrosine kinase inhibitor (TKI), lapatinib, an anti-HER2 therapy [55]. Within the GEO2R platform [62], the limma (Linear Models for Microarray Analysis) R package was used to analyze the GSE129254 expression array dataset [55] for DEGs. Table 2 denotes the MAF1 and TFIIIB mRNA expression increases in BT474 and SKBR3 HER2-positive breast cancer cells in response to anti-HER2 therapy. Our analysis shows that MAF1 expression increased in BT474 ($q = 0.026$) and SKBR3 ($q = 0.086$) cells treated with lapatinib. The only TFIIIB subunit differentially expressed in BT474 cells treated with lapatinib is BRF2 ($q = 0.078$). Volcano plots were generated to visualize the changes in MAF1 and TFIIIB expression for lapatinib treated BT474 (Fig 5A) and SKBR3 (Fig 5B) cells in the context of the entire genome [55]. We generated clustered heatmaps for the datasets analyzed, S1 Fig. Although microarray analysis informs DEGs, RNA sequencing provides increased specificity and sensitivity. Using the limma tool in the Galaxy platform [63], the raw sequencing data from the GSE158960 RNA sequencing dataset was analyzed from the control BT474 cells or BT474 cells treated with the humanized murine monoclonal antibody (MoAb) trastuzumab. MAF1 ($q = 4.51 \times 10^{-5}$), BRF2 ($q = 1.28 \times 10^{-6}$), and

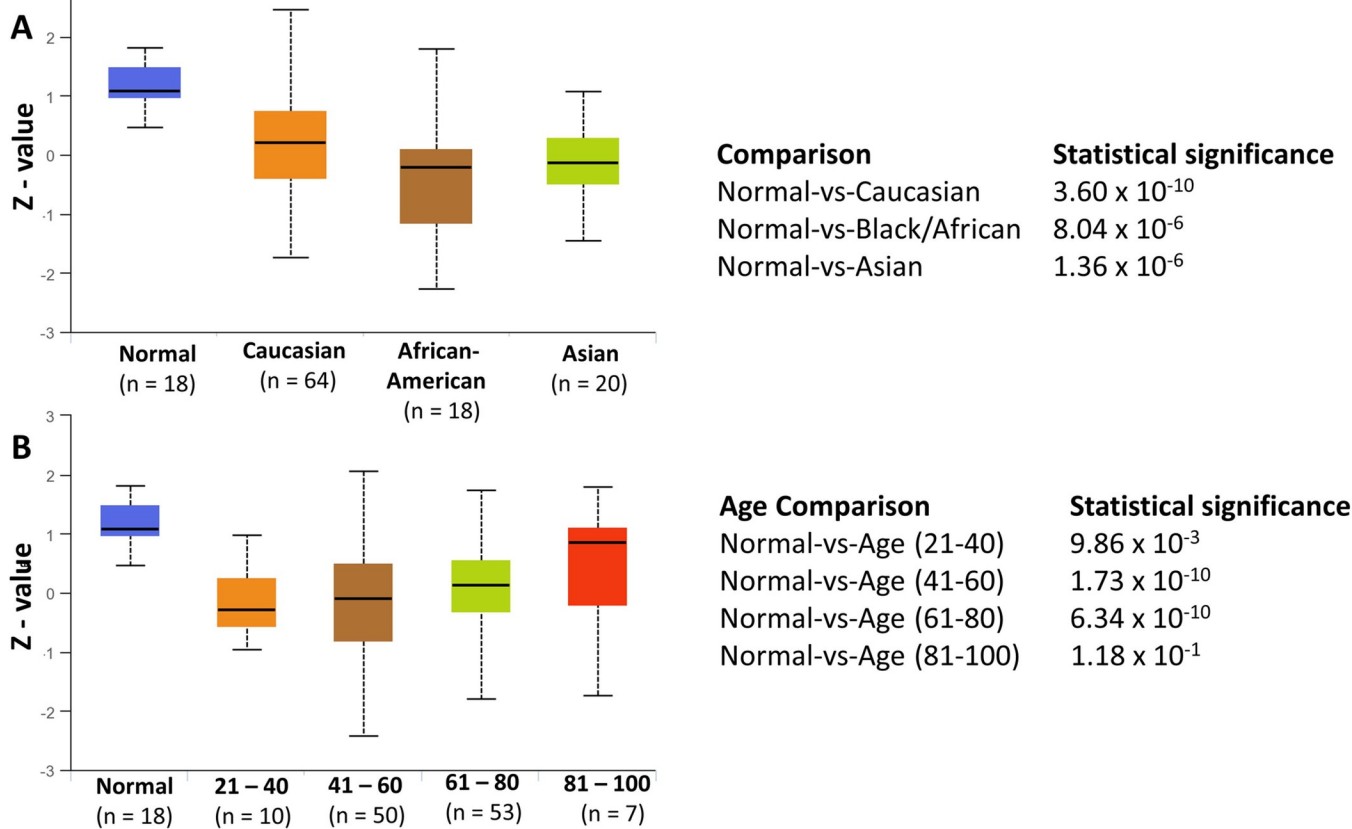

**Fig 4. MAF1 protein expression by patient race and age.** Using the University of Alabama at Birmingham Cancer data analysis portal (UALCAN) [60], we queried for MAF1 protein expression by race (**A**) and age (**B**) using data from Clinical Proteomic Tumor Analysis Consortium (CPTAC) and the International Cancer Proteogenome Consortium (ICPC) datasets [51]. The z-values presented are standard deviations of the median across samples, and log2 values were first normalized within each sample profile, then normalized across samples. Sample numbers are indicated; *p*-values are noted.

BDP1 ($q$ = 0.057) expression are increased in trastuzumab treated BT474 cells, Table 2. Next, we generated volcano plots to visualize the changes in MAF1 and TFIIIB expression for BT474 treated with trastuzumab in the context of the entire genome (Fig 5C) [56], and associated heatmaps are depicted in S1 Fig. MAF1, but not TFIIIB, expression is significantly increased by both anti-HER2 therapies tested in both HER2-positive cell lines analyzed. Table 2 and Fig 5 suggest that anti-HER2 therapies can potentially regulate MAF1 in HER2-positive breast cancer cell lines.

## MAF1 is predictive biomarker in HER2-positive breast cancer

Based on the results that anti-HER2 therapies increase MAF1 and TFIIIB expression in HER2-breast cancer cell lines (Table 2, Fig 5), we then queried breast cancer patients treated with anti-HER2 therapies using ROC Plotter [64]. ROC Plotter correlates transcriptomic gene expression data with cancer therapy response by integrating published gene expression data of 36 publicly available datasets with treatment data into a unified database to identify predictive biomarkers in breast cancer [64]. We queried the ROC Plotter [64] five-year relapse-free breast cancer patient dataset (n = 1,329), comparing samples from patients who did not relapse before five years versus patients who did relapse before five years. We analyzed this dataset for transcriptomic-level MAF1 expression in response to anti-HER2 therapies. Receiver operating characteristics (ROC) and Mann–Whitney tests compare gene expression and therapy

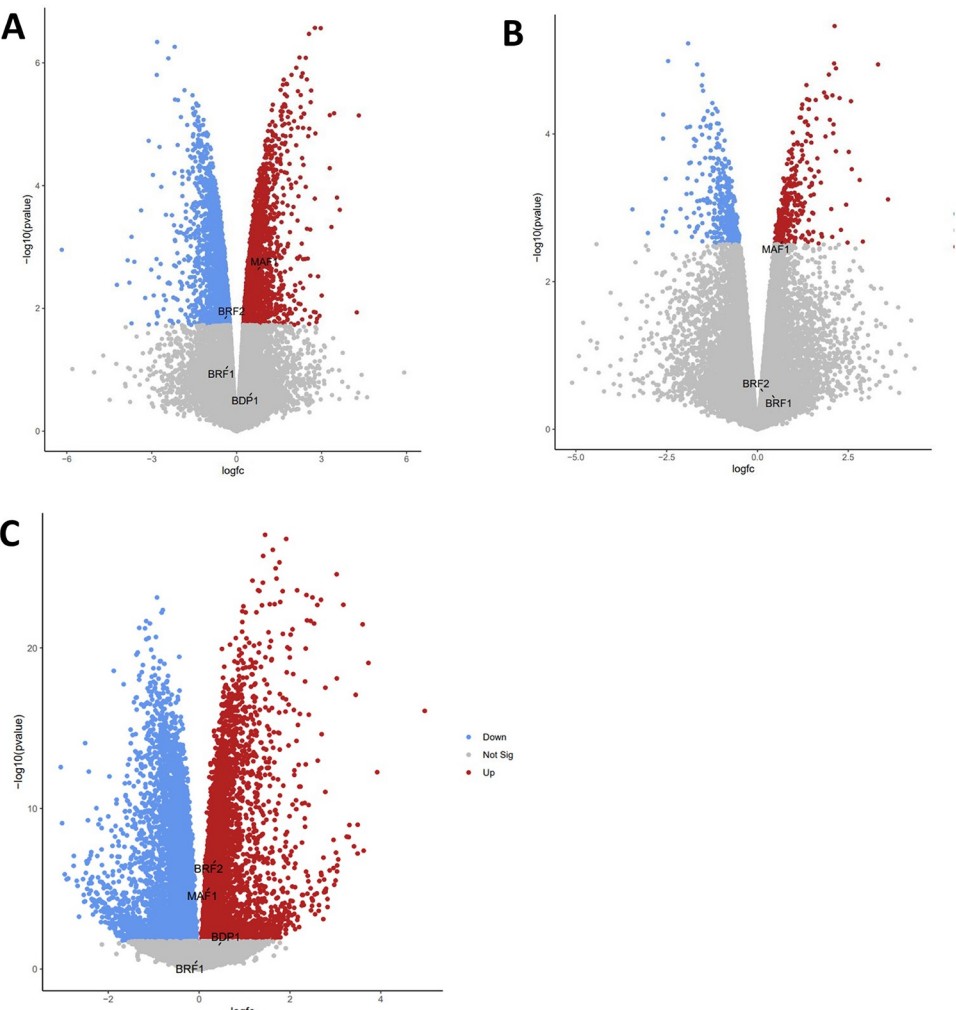

**Fig 5. MAF1 mRNA expression significantly increases in SKBR3 and BT474 breast cancer cell lines treated with anti-HER2 therapies.** We performed DEG analysis on publicly available microarray GSE129254 [55] and RNA sequencing GSE158969 [56] datasets. Anti-HER2 treatments are detailed in Table 2. The Galaxy platform [63] was used to generate volcano plots in ggplot2 to visualize MAF1 alterations with respect to all significantly altered genes in the datasets analyzed. (**A**) BT474 and (**B**) SKBR3 cells treated with lapatinib [55]. (**C**) BT474 cells treated with trastuzumab [56].

response [64]. We examined HER2, MAF1, and MYC expression in the five-year relapse-free breast cancer patient dataset, restricting our query to HER2-positive samples (n = 564) [64].

In Fig 6A, HER2 (gene symbol ERBB2; probe 216836_at*) expression increased (Mann-Whitney test *p*-value = 0.079) in response to trastuzumab treatment in HER2-positive breast cancer. As expected, HER2 is a predictive biomarker in HER2-positive breast cancer patients treated with trastuzumab (ROC *p*-value = $3.10 \times 10^{-2}$, AUC = 0.658) [64]. MAF1 expression (probe 222998_at*) increased (Mann-Whitney test *p*-value = 0.0012) in response to trastuzumab treatment in HER2-positive five-year relapse-free survival breast cancer patients (ROC *p*-value = $2.7 \times 10^{-6}$, AUC = 0.874), Fig 6B, suggesting MAF1 may be a predictive biomarker in HER2-positive breast cancer.

As demonstrated in Fig 1B, MAF1 and MYC alterations co-occur in breast cancer. MAF1 (Fig 2C) and MYC (Fig 2E) protein expression decreased in HER2-positive breast cancer. We

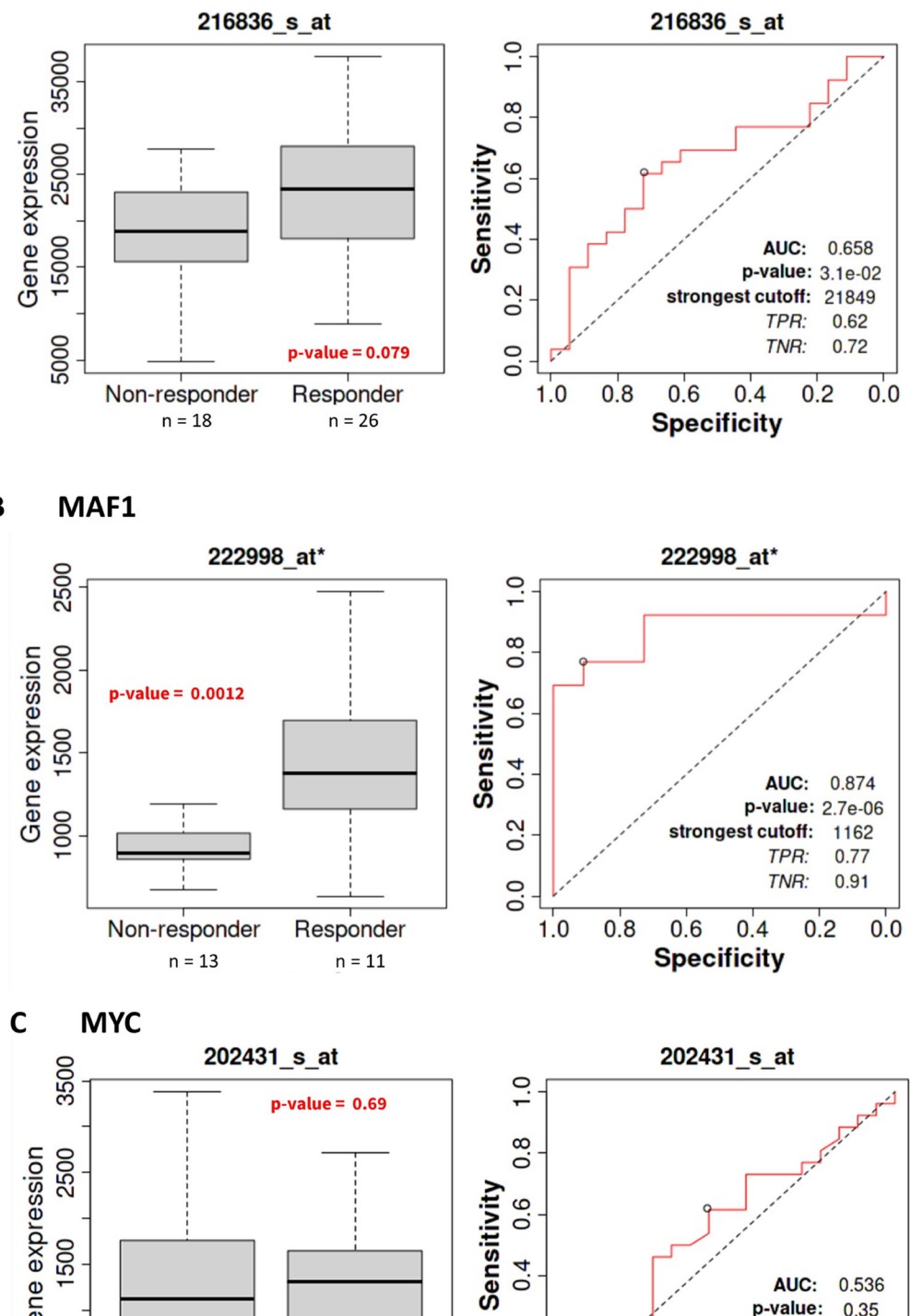

**Fig 6. Analysis of MAF1 as a candidate predictive biomarker in HER2-positive breast cancer.** ROC analysis [64] of (**A**) HER2 (gene symbol ERBB2; probe 216836_at*), (**B**) MAF1 (probe 222998_at*), and (**C**) MYC (probe 202431_at*) expression and specificity at five years relapse-free survival in response to trastuzumab treatment. Responders and nonresponders were compared using the Mann-Whitney test; p-values are provided. Expression fold change for HER2 is 1.2, MAF1 is 1.6, and MYC is 1.1. The area under the curve (AUC) and associated p-values are depicted. Only results with less than a 5% false discovery rate (FDR) are considered significant.

queried ROC Plotter [64] for breast cancer patients for transcriptomic-level MYC expression in response to anti-HER2 therapies. MYC expression (probe 202431_at*) was not altered (Mann-Whitney test p-value = 0.69) in response to trastuzumab treatment in HER2-positive breast cancer patients. Nor did we see a predictive association between MYC expression in response to trastuzumab in five-year relapse-free survival (ROC p-value = 0.35, AUC = 0.536), Fig 5C. The results in Fig 5C agree with a previously published study noting no change in MYC expression in response to trastuzumab [70].

## Discussion

The current study is the first to correlate MAF1 alterations with clinical outcomes in HER2-positive breast cancer. MAF1 amplification is frequent in invasive breast cancer and co-occurs with MYC (Fig 1). MAF1 expression decreased in subclasses of breast cancer, including HER2-positive breast cancer (Fig 2). Decreased MAF1 expression is associated with increased breast cancer stage (Fig 3). MAF1 mRNA expression is significantly increased in SKBR3 and BT474 breast cancer cell lines treated with anti-HER2 therapies (Table 2 and Fig 5). Furthermore, patients treated with trastuzumab had a significant increase in MAF1 expression ($p = 0.0012$) in the five-year relapse-free survival dataset ($p = 2.7 \times 10^{-6}$, AUC = 0.874), suggesting MAF1 can potentially be a predictive biomarker in HER2-positive breast cancer (Fig 6). Searching the TCGA [52, 53] dataset using the UALCAN platform [60] identified MAF1 ($p = 0.03$) as a prognostic marker in breast cancer.

HER2-positive breast cancers are known to develop resistance to anti-HER2 therapies, leaving patients with limited therapeutic options. Drug resistance to HER2-targeted therapy involves abnormal PI3K/AKT/mTOR or MAPK pathway activation downstream of HER2 [71]. Increased resistance to current anti-HER2 therapies has led to the development of novel therapies to treat HER2-positive breast cancer, including antibody-drug conjugate (ADCs), bispecific antibodies, chimeric antigen receptor T-cell therapy (CAR-T), immunotherapy, and nanotherapy [71]. This study demonstrates that the tyrosine kinase inhibitor (TKI) lapatinib and the HER2 targeting antibody trastuzumab increased MAF1 expression (Table 2, Figs 5 and 6).

We speculate that targeting MAF1 activity in HER2-breast cancer and HER2-positive breast cancer patients who have developed resistance to standard therapies will be of clinical value in assessing novel therapies. The monarcHER trial (NCT02675231), a phase 2 multi-center trial, aimed to compare the efficacy of the cyclin-dependent kinases 4 and 6 (CDK4/6) inhibitor abemaciclib plus the HER2 targeting antibody trastuzumab with or without the estrogen receptor antagonist fulvestrant to standard chemotherapy plus trastuzumab in women with advanced breast cancer [72]. Abemaciclib plus trastuzumab ± fulvestrant improved overall survival in women with HER2-positive breast cancer [73]. The monarcHER trial results prompted a preliminary investigation of MAF1 regulation by the CDK4/6 inhibitor abemaciclib in HER2-positive breast cancer. Using the edgeR tool in the Galaxy platform, we analyzed the raw sequencing data from the RNA sequencing datasets GSE157383 [57] and GSE99060 [58] from the MDA-MB-453 (HER2+), MDA-MB-453 (HER2-), and MDA-MB-361 (HER2+) breast cancer cells treated ± abemaciclib [63]. MAF1 is differentially expressed in

**Table 3. MAF1 and TFIIIB are differentially expressed in response to the CDK4/6 inhibitor abemaciclib.**

| GSE157383 | | | | |
|---|---|---|---|---|
| **Cell line** | **FDR** | **p-value** | **logFC** | **gene symbol** |
| MDA-MB-453 (HER2+) | $6.03 \times 10^{-10}$ | $1.18 \times 10^{-11}$ | 1.033473 | ERBB2 (HER2) |
| MDA-MB-453 (HER2+) | $1.19 \times 10^{-06}$ | $7.08 \times 10^{-08}$ | -0.59702 | MAF1 |
| MDA-MB-453 (HER2+) | 0.003564 | 0.000597 | 0.584456 | BRF2 |
| MDA-MB-453 (HER2+) | 0.066807 | 0.016799 | 0.192421 | BRF1 |
| MDA-MB-453 (HER2+) | 1 | 0.566101 | -0.26893 | MYC |
| MDA-MB-453 (HER2+) | 1 | 1 | 0 | BDP1 |
| MDA-MB-468 (HER2-) | 0.315 | 0.040 | -0.523 | ERBB2 (HER2) |
| MDA-MB-468 (HER2-) | 0.887 | 0.454 | -0.125 | MAF1 |
| MDA-MB-468 (HER2-) | 0.391 | 0.060 | 0.419 | BRF2 |
| MDA-MB-468 (HER2-) | 0.524 | 0.108 | 0.427 | BRF1 |
| MDA-MB-468 (HER2-) | 0.928 | 0.699 | 0.109 | MYC |
| MDA-MB-468 (HER2-) | 1 | 0.985 | -0.03 | BDP1 |
| GSE99060 | | | | |
| **Cell line** | **FDR** | **p-value** | **logFC** | **gene symbol** |
| MDA-MB-361 (HER2+) | 0.887 | 0.363 | 0.199 | ERBB2 (HER2) |
| MDA-MB-361 (HER2+) | 0.05 | 0.005 | -0.530 | MAF1 |
| MDA-MB-361 (HER2+) | 1 | 0.539 | 0.112 | BRF2 |
| MDA-MB-361 (HER2+) | 1 | 0.658 | -0.167 | BRF1 |
| MDA-MB-361 (HER2+) | 0.360 | 0.883 | 0.173 | MYC |
| MDA-MB-361 (HER2+) | 0.514 | 0.133 | -0.17 | BDP1 |

HER2-positive breast cancer cells ± abemaciclib but not in HER2-negative breast cancer cells, Table 3, and S1A Fig.

We then used the ROC Plotter platform with a dataset of drug-treated cell lines to determine if MAF1 is regulated by novel therapies newly developed to treat HER2-positive breast cancer [74]. Fekete and Győrffy created a combined dataset with chemosensitivity data of 1562 agents and transcriptome-level gene expression of 1250 cancer cell lines [74]. Drug Sensitivity datasets are derived from the Genomics of Drug Sensitivity in Cancer (GDSC) project [75], both GDSC1 and GDSC2 drug screening datasets, the Cancer Therapeutics Response Portal (CTRP) the version 2 drug screening dataset [76], and Cancer Dependency Map Consortium's DepMap portal were obtained from the PRISM Repurposing 19Q4 secondary screen dose–response dataset [77]. The lower tertile area under the dose-response curve (AUDRC) values were considered sensitive, and those in the upper tertile were considered resistant [74].

The Phase III ExteNET trial demonstrated that the irreversible pan-HER2 tyrosine kinase inhibitor neratinib, inhibiting PI3K/Akt and MAPK signaling, favorably increased disease-free survival in early-stage HER2-positive breast cancer patients, post-trastuzumab treatment [78]. ROC AUC and Mann-Whitney analyses demonstrate that MAF1 (AUC = 0.754, $p = 0.017$) altered expression in neratinib-treated HER2-positive breast cancer cell lines in the CTRP dataset, lower vs. upper tertile of AUDRC [74]. Table 4 presents the top breast cancer cell lines identified as sensitive or resistant to neratinib. Nine of the ten neratinib-sensitive breast cancer cell lines are HER2-positive, whereas nine of the ten neratinib-resistant breast cancer cell lines are HER2-negative. Our data suggest that MAF1 may be a novel target for therapeutic development for patients with HER2-positive breast cancer. More extensive clinical studies are warranted.

**Table 4. Top ten neratinib-treated sensitive (upper panel) and resistant (lower panel) CTRP breast cancer cell lines.**

| Sensitive Breast Cancer Cell Lines | HER2 Status | Standardized AUDRC |
|---|---|---|
| HCC2218 | + | 0.111 |
| ZR7530 | + | 0.144 |
| UACC812 | + | 0.17 |
| AU565 | + | 0.178 |
| SKBR3 | + | 0.189 |
| HCC1419 | + | 0.189 |
| BT474 | + | 0.192 |
| HCC202 | + | 0.24 |
| HCC1954 | + | 0.251 |
| CAL851 | - | 0.298 |
| **Resistant Breast Cancer Cell Lines** | | |
| HMC18 | - | 0.504 |
| CAMA1 | - | 0.496 |
| MCF7 | - | 0.471 |
| ZR751 | - | 0.46 |
| BT20 | - | 0.446 |
| MDAMB231 | - | 0.446 |
| HCC1428 | - | 0.443 |
| T47D | - | 0.439 |
| JIMT1 | + | 0.439 |
| HCC38 | - | 0.436 |

## Conclusions

TFIIIB-mediated RNA polymerase III transcription deregulation occurs in human cancers, including breast cancer [6–8, 28, 30, 79]. MAF1 is a negative regulator of TFIIIB-mediated RNA polymerase III transcription [42, 43] and has not been well characterized in breast cancer. MAF1 protein expression is significantly decreased in luminal, HER2-positive, and TNBC breast cancer subtypes and MAF1 mRNA expression is increased substantially in breast cancer cell lines treated with anti-HER2 therapies. Trastuzumab-treated HER2-positive breast cancer patients demonstrated an increased five-year relapse-free survival with significantly increased MAF1 expression. Further analysis shows that novel therapies in clinical trials for HER2 therapy-resistant breast cancers regulate MAF1 expression. Protein-protein interaction analysis, S2A Fig, indicates MAF1 interacts with known proteins deregulated in breast cancer. Our Kegg pathway analysis provides details for MAF1 involvement in a variety of human cancers, including breast cancers, S3 Fig. Together, data suggest MAF1 may serve as a predictive biomarker and a novel target for drug design for patients with breast cancer resistant to anti-HER2 therapies.

## Supporting information

**S1 Fig. Clustered heatmaps for GSE129254, GSE158969, GSE15783, and GSE99060 datasets.** Heatmaps were generated in Galaxy using the heatmap2 tool [63]. Top differentially expressed genes (DEGs) are presented. Heatmaps for GSE129254 lapatinib-DMSO SKBR3 (**A**) and BT474 (**B**), and GSE158969 Herceptin-control in BT474 cells (**C**) top DEGs associated with Table 2 analyses. Table 3 associated heatmaps for GSE157383 MBA-MB-453 (**D**), GSE157383 MDA-MB-468 (**E**), and GSE99060 MDA-MB-361 (**F**) breast cancer cells treated

with abemaciclib.
(TIF)

**S2 Fig. Network and enrichment analysis of MAF1.** (**A**) Using String 11.5 [80] we identified protein-protein interactions (PPI) for MAF1. The PPI enrichment p-value < 1.0 x 10–16 suggests indicates that the proteins may be biologically connected. (**B**) Table of protein interaction scores, interaction score > 0.4 was applied.
(TIF)

**S3 Fig. KEGG pathway analysis for MAF1.** The most significant (FDR) Kegg pathways involved in the MAF1 network are presented.
(TIF)

## Author Contributions

**Conceptualization:** Laura Schramm.

**Data curation:** Stephanie Cabarcas-Petroski, Gabriella Olshefsky, Laura Schramm.

**Formal analysis:** Stephanie Cabarcas-Petroski, Gabriella Olshefsky, Laura Schramm.

**Investigation:** Stephanie Cabarcas-Petroski, Laura Schramm.

**Methodology:** Stephanie Cabarcas-Petroski, Gabriella Olshefsky, Laura Schramm.

**Validation:** Stephanie Cabarcas-Petroski, Laura Schramm.

**Writing – original draft:** Laura Schramm.

**Writing – review & editing:** Stephanie Cabarcas-Petroski.

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
