## [Editor Report · Decision Letter 0]

20 Jun 2023

PONE-D-23-17750MAF1 is a Predictive Biomarker in HER2 Positive Breast CancerPLOS ONE

Dear Dr. Cabarcas-Petroski,

Thank you for submitting your manuscript to PLOS ONE. After careful consideration, we feel that it has merit but does not fully meet PLOS ONE’s publication criteria as it currently stands. Therefore, we invite you to submit a revised version of the manuscript that addresses the points raised during the review process.

We look forward to receiving your revised manuscript.

Kind regards,

Divijendra Natha Reddy Sirigiri

Academic Editor

PLOS ONE

Journal Requirements:

https://pubmed.ncbi.nlm.nih.gov/35406430/

In your revision ensure you cite all your sources (including your own works), and quote or rephrase any duplicated text outside the methods section. Further consideration is dependent on these concerns being addressed.

6. Please remove your figures from within your manuscript file, leaving only the individual TIFF/EPS image files, uploaded separately. These will be automatically included in the reviewers’ PDF.

Additional Editor Comments:

Request authors to make figures with more clarity.

Graphs fonts on the axis can not be seen (rather fuzzy in the current version, Fig 1-4). May be text on the axis can be retyped. Please recheck the typos.

---

## [Author Response · Author response to Decision Letter 0]

27 Jul 2023

Dear Dr. Divijendra Natha Reddy Sirigiri,

We sincerely thank you and the reviewers for a comprehensive review and an opportunity to respond to each point raised by the academic editor and reviewer(s).

Below we reply to each point raised:

Reply: To address the formatting issues, please note that we first converted our manuscript to the PLOS ONE guidelines using the website and PDFs provided above. Next, we turned on tracked changes to address text changes. Our original submission was in a format significantly different than PLOS ONE. We provide both the required ‘Revised Manuscript with tracked changes’ and a second copy accepting the changes titled ‘Manuscript.”

2. We noticed you have some minor occurrence of overlapping text with the following previous publication(s), which needs to be addressed: https://pubmed.ncbi.nlm.nih.gov/35406430/

In your revision ensure you cite all your sources (including your own works), and quote or rephrase any duplicated text outside the methods section. Further consideration is dependent on these concerns being addressed.

Reply: We thank the reviewers for this comment. In the manuscript link provided above, we examined the role for the TFIIIB subunit BDP1 in breast cancer; BDP1 had never been studied before. Our current study examines the role of MAF1, a negative regulator of RNA polymerase III transcription, in breast cancer. We used similar bioinformatics approaches and expanded the number of datasets analyzed. We have analyzed our current submission via Turnitin, Grammarly, and WORD Editor, and none of the similarity analyses did not return text similarity between the two manuscripts. We updated the methods section to include a statement to refer the reader to the above manuscript link to examine the methods used. If the editorial office provides a copy of the similarity report, we will happily address any text similarities.

Reply: Thank you. All dataset accession numbers are provided in the manuscript. We have used public datasets. Table 1 summarizes the public datasets used, including accession numbers.

Reply: We thank the reviewers. The “data not shown” data are not a core part of the presented research, and we have removed the phrase that refers to these data. 

Reply: Thank you. We have included a full ethics statement in our methods section. This study did not require IRB or ethics committee approval as we utilized in silico analysis for publicly available datasets. We did not collect human samples (tissues, cells, fluids) or communicate with human participants. The data used for analysis were retrieved from publicly accessible datasets. The accession numbers for all datasets can be found in Table 1. 

6. Please remove your figures from within your manuscript file, leaving only the individual TIFF/EPS image files, uploaded separately. These will be automatically included in the reviewers’ PDF.

Reply: Thank you. We removed the figures from the body of the manuscript when we revised the manuscript in PLOS ONE formatting. 

Reply: We thank the reviewers. We have correctly formatted the manuscript, moved support captions to the end of the document, and updated the in-text citations. 

Reply: Thank you. We have checked that none of the references cited have been retracted.

Additional Editor Comments:

Request authors to make figures with more clarity.

Graphs fonts on the axis can not be seen (rather fuzzy in the current version, Fig 1-4). May be text on the axis can be retyped. Please recheck the typos.

Reply: Thank you. We have retyped the axis text for Figures 1 – 4.

Lastly, we have removed the funding statement from the manuscript as requested by Audyssa Banlaygas Auditor, PLOS ONE, per PLOS ONE guildelines. As this research did not receive external funding and as we cannot choose "St. John's University" as a funding source within the submission system, we respectfully request that per the recommendation of the auditor, the online submission form be updated on our behalf to state:

"We acknowledge St. John's University for funding our study."

Thank you.

---

## [Editor Report · Decision Letter 1]

1 Sep 2023

MAF1 is a Predictive Biomarker in HER2 Positive Breast Cancer

PONE-D-23-17750R1

Dear Dr. Cabarcas-Petroski,

We’re pleased to inform you that your manuscript has been judged scientifically suitable for publication and will be formally accepted for publication once it meets all outstanding technical requirements.

Kind regards,

Divijendra Natha Reddy Sirigiri

Academic Editor

PLOS ONE
---

## [Editor Report · Acceptance letter]

28 Sep 2023

PONE-D-23-17750R1 

MAF1 is a Predictive Biomarker in HER2
Positive Breast Cancer 

Dear Dr. Cabarcas-Petroski:

I'm pleased to inform you that your manuscript has been deemed suitable for publication in PLOS ONE. Congratulations! Your manuscript is now with our production department. 

Kind regards, 

on behalf of

Dr. Divijendra Natha Reddy Sirigiri 

Academic Editor

PLOS ONE